# Body Fluid-Derived Stem Cells: Powering Innovative, Less-Invasive Cell Therapies

**DOI:** 10.3390/ijms26094382

**Published:** 2025-05-05

**Authors:** Adam David Goff, Xinyue Zhang, Biju Thomas, Sally Shin Yee Ong, Anthony Atala, Yuanyuan Zhang

**Affiliations:** 1Wake Forest Institute for Regenerative Medicine, School of Medicine, Wake Forest University, Winston-Salem, NC 27101, USA; agoff@wakehealth.edu (A.D.G.); xyzhang179@gmail.com (X.Z.);; 2School of Medicine, Wake Forest University, Winston-Salem, NC 27101, USA; 3Keck School of Medicine of USC, University of Southern California, Los Angeles, CA 90033, USA; 4Department of Ophthalmology, School of Medicine, Wake Forest University, Winston-Salem, NC 27101, USA

**Keywords:** body fluid-derived stem cells, stem cell therapy, regeneration, personalized medicine

## Abstract

Stem cell therapy offers significant promise for tissue regeneration and repair. Traditionally, bone marrow- and adipose-derived stem cells have served as primary sources, but their clinical use is limited by invasiveness and low cell yield. This review focuses on body fluid-derived stem cells as an emerging, non-invasive, and readily accessible alternative. We examine stem cells isolated from amniotic fluid, peripheral blood, cord blood, menstrual fluid, urine, synovial fluid, breast milk, and cerebrospinal fluid, highlighting their unique biological properties and therapeutic potential. By comparing their characteristics and barriers to clinical translation, we propose body fluid-derived stem cells as a promising source for regenerative applications, with continued research needed to fully achieve their clinical utility.

## 1. Introduction

Regenerative medicine seeks to restore the structure and function of damaged tissues and organs by harnessing the body’s intrinsic healing mechanisms. Central strategies in this field include tissue engineering, extracellular vesicle (EV)-based therapies, and cellular approaches—all of which depend on effectively harvesting a reliable source of stem cells for their ability to self-renew, differentiate, and secrete bioactive factors with immunomodulatory and trophic effects [1,2,3].

Traditionally, stem cells have been harvested from solid tissues such as bone marrow and adipose tissue. While effective, these sources present several limitations. Collection procedures are invasive, often requiring techniques like drilling into the iliac crest or performing liposuction, followed by complex processing methods such as enzymatic digestion or mechanical centrifugation to isolate stem cells from surrounding extracellular matrices [4]. A wide range of isolation techniques are also utilized, which can lead to variable concentrations and differentiation potentials of the harvested stem cells from solid tissue [5,6]. This complexity can result in variable cell yields and differentiation potentials [5,6]. Furthermore, stem cell quantity and potency decline with age, reducing their therapeutic viability in older individuals [7,8,9].

Given these challenges, interest has grown in less-invasive alternatives—specifically, body fluid-derived stem cells (BFSCs). Our recent research in urogynecology supports the use of BFSCs as a promising, accessible source with regenerative potential across multiple organ systems [10]. Their comparative ease of harvest and accessibility make them attractive candidates for standardized and widely used sources of stem cells for therapeutic use. However, specific sources of BFSCs with known stem cell populations can have unique characteristics that could influence their potential therapeutic application, which will be discussed further in this review.

Body fluids like umbilical cord blood, amniotic fluid, urine, breast milk, menstrual blood, peripheral blood, synovial fluid, and cerebrospinal fluid (CSF) are essential for physiological homeostasis. Beyond their primary roles, these fluids may offer a viable, less invasive source of stem cells for regenerative applications.

This review examines the characteristics of BFSCs across various body fluids, outlines current isolation methods, and evaluates their regenerative potential in tissue repair. We also consider the challenges and future directions of integrating BFSCs into clinical practice.

## 2. Classifying Stem Cells

Scientists have developed many classifications of stem cells based on various defining characteristics. Understanding these classification methods, such as potency, developmental origin, or differentiation potential, is essential to ensure clarity and consistency when discussing current regenerative medicine advances, which can rely heavily on stem cells and their unique properties.

## 3. Body Fluid-Derived Stem Cells

Several body fluids harbor distinct stem cell populations that offer unique advantages over those from solid tissues. This section highlights promising candidates that can be obtained through minimally or non-invasive methods—amniotic fluid, peripheral blood, umbilical cord blood, menstrual fluid, urine, synovial fluid, cerebrospinal fluid (CSF), and breast milk (Table 1). These fluids can be broadly categorized as being retained (e.g., peripheral blood stem cells [PBSCs] in blood, collected in a minimally invasive manner) or excreted (e.g., breast milk-derived mesenchymal stem cells [BMSCs] in breast milk, collected in a non-invasive manner). These sources are key to therapeutic stem cell applications due to their inherent sterility or the low contamination risk associated with their collection (Figure 1).

In contrast, while certain body fluids like gastrointestinal mucus, saliva, tears, sweat, and vaginal fluid contain cells, they are less suitable for cell-based therapies compared to the aforementioned sources. Their constant exposure to the external environment and resident microbial flora elevate the risk of pathogen contamination, potentially impairing stem cell viability, potency, and immunological properties [29], as well as posing a risk of infection [30] that can compromise cell cultures. Furthermore, although some other body fluids are sterile, their collection often presents significant accessibility challenges or involves dangerous procedures, making cells derived from fluids like the abdominal cavity, thoracic cavity, or pericardium less ideal for therapeutic applications due to potential complications.

### 3.1. Fetal or Neonatal Sources

#### 3.1.1. Amniotic Fluid-Derived Stem Cells

Amniotic fluid-derived stem cells (AFSCs) are a subset of cells within a heterogeneous population from the fetus and the amniotic membrane [31]. These cells can come from the developing embryo’s three germ layers; therefore, AFSCs exhibit multilineage differentiation potential. AFSCs are collected during routine amniocentesis; however, only a small subset of cells from this fluid display stem cell properties. After amniocentesis, these cells are commonly isolated on a plastic adherence culture and immunoselection is subsequently utilized, targeting the c-kit (CD117) surface antigen, which is indicative of stem cell properties [32].

AFSCs exhibit a unique combination of properties, positioning them as a promising resource in regenerative medicine. These cells express epigenetic markers similar to MSCs, which include CD44, CD73, CD90, and CD105. Interestingly, MSC-like stem cells derived from amniotic fluid demonstrated lower senescence rates and more rapid proliferation during culture expansion than MSCs derived from bone marrow, further supporting their therapeutic use [33,34]. Epigenetic mechanisms like DNA methylation and histone modification, which are proposed to influence the differentiation and therapeutic potential of AFSCs, have been investigated but not fully characterized. Recent advances have highlighted the importance of microRNAs (miRNAs) in influencing the WNT, MAPK, and TGF-*β* signaling pathways that affect AFSC differentiation [35]. In addition, AFSCs from an unknown tissue origin displayed superior multipotency, proliferation, and lower rates of senescence when compared to AFSCs of known tissue origin (kidney- and lung-specific) [36]. This addresses the translational barrier of heterogeneity; however, the composition of the AFSC population continues to be debated.

While AFSCs demonstrate promising differentiation capabilities in vitro, translating this potential into effective in vivo therapies remains an active area of research. Due to the heterogeneous population, amniotic fluid is a source of diverse stem cell types, including those originating from the skin, gastrointestinal, urinary, and respiratory systems [31]. This heterogeneity necessitates specific isolation techniques to obtain targeted cell populations for differentiation. Furthermore, the differentiation potential of these stem cells can vary depending on the gestational age of the amniotic fluid at the time of collection [37] (Table 2).

#### 3.1.2. Umbilical Cord Blood-Derived Stem Cells 

Umbilical cord blood is a readily accessible source of postnatal stem cells that is routinely collected from the umbilical cord and placenta after birth without compromising fetal well-being, typically yielding volumes ranging from 40 mL to 200 mL. Umbilical cord blood is rich in two main types of stem cells—hematopoietic stem cells (HSCs) [68] and mesenchymal stromal cells (MSCs) [69]. HSCs have the capacity to differentiate into all mature blood cell types, while MSCs have been successfully induced to different types of cells for tissue repair [69]. This positions umbilical cord blood as a promising and ethically sound alternative to bone marrow MSCs have been used for various regenerative medicine applications, particularly in the field of tissue engineering. Similarly to AFSCs, miRNA is a growing field of interest that has been shown to be applicable in the regenerative medicine application of USCs. A particular miRNA sequence of interest (miR-21) promotes the regeneration of bone and epithelial tissue through the interaction with the PTEN/PI3K/Akt pathway when delivered within umbilical cord blood-derived stem cell (UCBSC) exosomes to diseased tissue [70]. This is just one of many valuable miRNAs that are involved in tissue regeneration using UCBSCs. The manipulation of the Notch signaling pathway has also been shown to provide an avenue for the targeted therapy of UCBSCs. For example, UBCSCs can be induced to undergo differentiation into myocardial cells when treated with 5-azacytidine (5-azac), which is proposed to alter UBCSCs through the Notch signaling pathway [71].

Although most viable stem cells obtained from cord blood are hematopoietic, the concentration of MSCs derived from cord blood is typically lower [44]. Notably, UCB-MSCs demonstrate higher proliferative potential and a preserved differentiation capacity compared to bone marrow MSCs [42], yet their scarcity in cord blood remains a significant limitation (Table 2). Ongoing research is crucial to confirm that these cells retain their differentiation potential and therapeutic properties throughout the expansion and cell banking that are needed to produce therapeutic doses.

### 3.2. Adult Sources

#### 3.2.1. Peripheral Blood-Derived Stem Cells (PBSCs)

HSCs, which are characterized by the CD34 surface marker, are the most commonly used subtype of PBSCs in clinical settings. Other populations of PBSCs, which are defined by various surface markers, have been discovered; however, their expression patterns and characteristics are not as explicitly defined, leading to differences in isolated PBSC populations [72]. This wide variability in the characterization of PBSCs requires the future standardization of the characterization of PBSCs to deliver a reliable population of PBSCs [73].

Current research has demonstrated that PBSCs have a breadth of utility across numerous regenerative medicine applications. Some of these advancements include extracellular vesicles (EVs) that contain genomic and proteomic elements, which interact with the EGR1 gene to influence PBSC regulation [74]. In addition, the proliferative properties and clinical applications of PBSCs are currently being optimized in relation to multiple applications, including biomaterials and co-culture systems [75].

Compared to BM, peripheral blood contains a lower concentration of MSCs. However, the mobilization of MSCs from BM into peripheral blood is feasible through agents like granulocyte colony-stimulating factor (G-CSF) and filgrastim [76]. This increases the therapeutic potency of peripheral blood while eliminating the need for invasive procedures like BM aspiration to collect MSCs. HSCs from peripheral blood are less concentrated than UCBSCs but offer a more feasible collection method through apheresis after growth factor injection; however, obtaining a therapeutic level of cells still remains a significant challenge for therapeutic use of these BFSCs [46,47,48]. In addition, PBSCs had a higher incidence of chronic graft vs. host diseases (GVHDs) than HSCs from BM (Table 2), but did display a faster neutrophilic and platelet engraftment [49]. This can reduce the duration of the aplastic phase after induction chemotherapy to minimize adverse events like opportunistic infections and bleeding complications. In addition, the rapid advancements in the characterization and application of PBSCs have exposed a discrepancy in regulatory oversight to ensure safe practices in the clinical application and sterile ex vivo expansion of these cells [77].

#### 3.2.2. Menstrual Blood-Derived Stem Cells (MenSCs)

MenSCs express markers that have previously been identified as being specific to mesenchymal stem cells [50,78]. This expression pattern and their anti-inflammatory and regenerative properties make them a promising option for cell-based therapies [79]. MenSCs also have enhanced angiogenic potential compared to BMSCs because of their additional expression of PDGF-B and their superior performance within sprouting and migration assays [80]. In addition, novel isolation techniques using magnetic bead sorting using the SUSD2 surface marker have made the isolation of these stem cells more feasible compared to previously established cell sorting techniques [52]. IL-6 has also been investigated as a potential adjunct to enhance the self-renewal of MenSCs utilizing the WNT/*β*-catenin pathway [81].

These MSCs have been shown to differentiate into numerous cell types and offer a non-invasive alternative to MSCs harvested from bone marrow. Despite current advances, there is still a need to generate an optimal and reproducible protocol to isolate, expand, and store stem cells from this fluid [50]. Various properties may be affected by donor age, hormonal status, and contraceptive use. The sterility of the sample requires careful handling (Table 2). Their replicative ability may be lower compared to that from other MSC sources like adipose tissue. The potential tumorigenicity and genomic instability of MenSCs is also still being investigated, limiting their clinical translation [82].

#### 3.2.3. Urine-Derived Stem Cells (USCs)

USCs were first successfully isolated and serially expanded in 2008 using a selective medium. The study demonstrated that USCs, which are present in small quantities in voided urine, expressed progenitor-like features that offered a novel source of stem cells that are capable of clinical regenerative application [57]. These cells are harvested non-invasively and likely originate from the kidney mesoderm or nephron tubules [83]. Furthermore, they show similarities to mesenchymal stem cells through their proliferative, migratory, and differentiation capacities [55]. They can also differentiate into numerous other cell types, such as neuronal, endothelial, and smooth muscle cells. Another advantage of USCs is that they have a high telomerase activity, conserve their karyotype throughout in vitro expansion, and have immunomodulating properties [56]. Qiqao et al. also found that Bodine, which is a small-molecule compound, can preserve the differentiation capacity and promote the proliferation of USCs through the upregulation of the WNT/*β*-catenin pathway, which addresses current limitations in relation to the clinical application of USCs [84].

Generating a large and therapeutic population of cells from this source remains challenging, and research is ongoing to explore their potential in regenerative medicine, cell therapy, and drug discovery [85,86]. Their ease of harvest, low cost, preserved characteristics after subsequent passages, and wide range of therapeutic applications make them an attractive source of stem cells for therapeutic use in regenerative medicine. In addition, the tumorigenicity of USCs was assessed using an in vivo model. This study found that USCs are not tumorigenic even after using cell culture conditions that increase the level of therapeutic factors [87]. Ultimately, the further standardization of isolation techniques is required for USCs to be used widely in clinical settings. Despite advancements in their application, USC use still faces limitations. There is potential for contamination, particularly in collecting urine samples from females. In addition, their isolation efficiency can vary significantly between individuals. Despite successful differentiation into various cell types, the potential for certain lineages, such as chondrogenic, is lower compared to that of other MSC sources. Furthermore, the regenerative ability of USCs may be reduced in stem cells derived from aged donors or patients with certain diseases like diabetes (see Table 2).

#### 3.2.4. Synovial Fluid-Derived Stem Cells (SFSCs)

Synoviocytes produce synovial fluid and reside within the joint capsule to reduce friction, absorb compressive forces, and provide nutrients to the surrounding avascular tissue, such as cartilage. In addition, this fluid contains MSCs that are proposed to be mobilized from the surrounding synovium, in a process that is mediated by transforming growth factor beta-3. Other origins of these cells are also considered, like bone and cartilage [59,88]. The synovial fluid can then be aspirated and the SFSCs can be expanded using adherent or suspended cell culture methods. Furthermore, the stem cells harvested from the synovial fluid were shown to have a lower proliferative potential than those isolated from the synovial membrane [61].

In the setting of OA and other joint trauma, the number of MSCs within the synovium is increased compared to that of healthy tissue, indicating the body’s own endogenous protective and healing response through the increased expression of TGF-*β*, as well as superoxide dismutase activity. Unfortunately, the number of stem cells is limited, allowing for the inevitable progression of this disease [89]. The ability to differentiate into cartilage cells exerts anti-inflammatory effects, and their immunomodulatory potential makes them promising candidates for osteoarthritis treatment [60]. However, the amount of synovial fluid available, and therefore the amount of SFSCs available, is limited. In addition, aspirating synovial fluid is moderately invasive and requires an advanced technique. Their quality and quantity can be affected by age and joint disease (Table 2). New strategies are being developed to minimize the need for cell culture expansion. An example is stem cell mobilizing devices (STEMs) that agitate the synovium, which have been shown to significantly increase the amount of SFSCs collected [90].

#### 3.2.5. Breast Milk-Derived Stem Cells (BmSCs)

The intrinsic properties of breast milk are now widely known to be beneficial to nursing infants. Breast milk contains an irreplaceable composition of macronutrients, micronutrients, immune cells, growth factors, immunomodulators, and prebiotics [91,92]. Of particular interest to this review is its containment of stem cells. Present in breast milk, these cells show potential for differentiating into various cell types from all three germinal layers, including mammary epithelial cells, adipocytes, and neurons [64,65]. hBmSCs could also be isolated and cultured in vitro after extracting this fluid using a breast pump, which ensures the non-invasive advantage of body fluid-derived stem cells. In addition, one study found that BmSCs expressed genes found within ESCs, allowing them to potentially serve as an alternative to circumvent the ethical limitations of harvesting ESCs [63]. Interestingly, breast milk also has the highest concentration of miRNA among all body fluids [93]. This suggests their critical role in offspring development through epigenetic modulation. miRNA can also cross the blood barrier to contribute to neurocognitive development and reduce gastrointestinal inflammation [94].

Due to the complex characterization of BmSCs, isolating a homogenous population for therapeutic use that will have conserved benefits from each individual is challenging. The heterogeneity of this stem cell population is also influenced by maternal diseases such as diabetes [94]. In addition, breast milk produces a low yield of BmSCs, and optimal cell culture conditions have not yet been identified [64,65]. Cell yield may vary between individuals and lactation stages (Tabel 2).

#### 3.2.6. Cerebrospinal Fluid-Derived Neural Stem Cells

Previous research has shown that one of the locations in the brain that includes NSCs is the CSF, which can be obtained through lumbar puncture in fetal patients [66]. In this study, CSF was aspirated from myelomeningoceles, and neural progenitor cells were further isolated and differentiated in vitro into cells resembling neurons, astrocytes, and oligodendrocytes. This is revolutionary as it highlights a potential avenue to access NSCs in fetal patients through a minimally invasive technique, supporting the consistent advantages of body fluid stem cells. Despite the inability to produce stem cells, CSF is near the ventricular–subventricular zone, which is proposed to participate in neurogenesis and be involved in the production of NSCs [95]. NSCs have also been successfully isolated from cerebrospinal fluid-touching neurons in the central canal of neonatal mice [96]. A recent study showed that this body fluid, which is produced by the choroid plexus, promotes the proliferation, survival, and neuronal differentiation of BM-MSCs and NSCs through its microenvironment and cell signaling [97].

There has been variability in research outcomes involving NSCs. Thus, further investigation is needed to assess the quality and therapeutic capacity of NSCs originating from various neuronal locations and pathologies. In addition, the low cell yield of CSF and low viability make analyzing and utilizing them in a therapeutic manner very challenging (Table 2). To address this clinical barrier, Singh et al. developed a cryopreservation method that permits the long-term storage of cells originating from the CSF. However, this is a very novel method, and the large-scale production and standardization of storage protocols are necessary [98].

While CSF holds promise as a source of NSCs or signaling molecules that are relevant to therapies, its routine isolation faces significant limitations due to the invasive nature of lumbar puncture. This procedure carries inherent risks (e.g., post-dural puncture headache and infection), which demand specialized technical expertise for both collection and subsequent processing in order to maintain sample integrity. These technical complexities, coupled with substantial ethical considerations regarding patient burden, particularly in non-critical research, constrain widespread application. However, given the difficulties in obtaining autologous NSCs from less-invasive sources, CSF remains a compelling alternative. Its proximity to the central nervous system suggests it may contain unique endogenous neural progenitors and a rich array of signaling molecules that are directly relevant to neural physiology and pathology, potentially offering autologous cell sources or novel therapeutic targets. Future research must address the technical challenges of sample handling, prioritize less-invasive access methods, and conduct a careful ethical evaluation to fully realize CSF’s therapeutic potential while minimizing the risk of lumbar puncture and patient burden.

The landscape of iPSC generation has evolved significantly, with a notable shift towards the utilization of more readily accessible cell sources such as blood [99] and urine [85,100,101,102]. These cell types have increasingly become the preferred starting material for reprogramming into induced pluripotent stem cells, offering a clear advantage over traditional skin fibroblasts. This preference stems primarily from the non-invasive nature of sample collection for both blood draws and urine collection, contrasting sharply with the more invasive biopsy procedures required to obtain skin fibroblasts. This less-invasive approach enhances donor comfort, reduces potential risks, and facilitates broader participation in research studies and potential therapeutic applications.

Research on BFSCs remains in the early stages; thus, their full therapeutic potential and safety profiles require further investigation. To enhance the clinical utility of BFSCs, their continued optimization and careful evaluation is required. Given the variability among BFSC sources, refining key parameters—such as dosage, administration route, and treatment frequency—is critical. Preclinical studies in large animal models can be used to evaluate safety, efficacy, and long-term outcomes before human trials. Priorities include (a) dosage optimization to maximize efficacy while minimizing adverse effects; (b) identifying the most effective delivery method (e.g., intravenous, intramuscular, or localized injection); (c) determining optimal treatment frequency to sustain therapeutic benefits; and (d) conducting rigorous preclinical testing. Although significant challenges remain, the unique properties of BFSCs present a promising foundation for future medical applications.

## 4. Potential Applications of Body Fluid-Derived Stem Cells

BFSCs have shown therapeutic potential in a variety of tissue and organ systems, as summarized in (Table 3).

### 4.1. Musculoskeletal System (Bone, Cartilage, Tendons, and Ligaments)

The regeneration of bone, cartilage, tendons, and ligaments is a significant area of focus for BFSC research. Several sources show promise. BFSCs, such as synovial fluid and peripheral blood, carry a lower risk of morbidity and adverse effects than bone marrow and adipose tissue, while maintaining a similar regenerative potential [61,103]. Notably, PBSCs exhibit some promise in bone and cartilage repair. In preclinical studies, AFSCs have also demonstrated potential in repairing bone defects and cartilage lesions, which are essential, as accessible sources of stem cells are scarce in fetuses [104,105].

MenSCs have shown promise in preclinical bone and cartilage repair studies. This source of stem cells has been demonstrated to be a safer and more effective alternative to BMMSCs, exhibiting lower tumorigenicity and a higher proliferative capacity [106,107]. USCs have also demonstrated potential in bone and cartilage repair research [108]. One study has shown that USCs cannot be differentiated into osteocytes as effectively; however, they had a higher proliferation potential [109]. Further research is being conducted to enhance their osteogenic differentiation potential [110]. Finally, SFSCs, with their high chondrogenic potential, are particularly relevant for treating osteoarthritis and cartilage injuries, showing potential in bone repair and tendon/ligament repair [111]. As the world population is rapidly aging, the prevalence of degenerative diseases like osteoarthritis (OA) is increasing. OA cases have risen by 113.35% between 1190 and 2019 [154]. Novel regenerative therapeutics, like synovial fluid-derived stem cells, can offer a potential solution to this significant disease burden. To treat OA, MSCs from the synovial fluid can be harvested endogenously or exogenously and then injected into the affected joint.

### 4.2. Cardiovascular System (Heart and Blood Vessels)

Cardiovascular disease (CVD), especially ischemic heart disease, is the leading cause of death worldwide, and the number of cases of CVD has nearly doubled between 1990 and 2019 [155]. This primarily affects the elderly population, but congenital heart disease (CHD) also impacts global health, as an estimated 8 in every 1000 newborns are diagnosed with CHD [156]. To minimize this burden and reduce the need for other invasive treatments like cardiac transplants, cardiac regeneration and vascular repair using stem cells are critical for treating cardiovascular diseases. AFSCs have shown potential in myocardial regeneration in many therapeutic applications like myocardial infarction and valve repair in children [112].

This source of stem cells is desirable for treating CHD as AFSCs are extracorporeal, which minimizes the risk of injuring the newborn. Bollini et al. also demonstrated that after myocardial infarction in a rat model, the injection of AFSCs reduces the size of the infarction and the number of dead myocytes [113]. PBSCs, specifically MSCs, have demonstrated potential in cardiac regeneration and have been applied to promote angiogenesis, vascular repair, and reduce fibrosis in order to ultimately improve cardiac function [114]. UCBSCs and MenSCs have shown potential in myocardial repair through direct cardiomyocyte regeneration, paracrine effects, and immunomodulation [115,116].

### 4.3. Hematopoietic System (Blood and Bone Marrow)

The regeneration of the hematopoietic system is essential for treating hematological malignancies and disorders. UCBSCs are widely used in transplantation to treat hematological disorders due to their rich source of HSCs and lower risk of GVHDs. These cells are used mainly in allogenic hematopoietic blood cell transfusions to treat diseases like leukemias, lymphomas, aplastic anemia, thalassemia, and diabetes [117]. In addition, the hematopoietic stem cells isolated from cord blood are relatively immune compared to those isolated from bone marrow and peripheral blood. Therefore, this characteristic allows them to tolerate a higher disparity in human leukocyte antigen (HLA) matching. It significantly reduces the probability of developing GVHD during an allogeneic transplant [43]. PBSCs, containing HSCs, are primarily used in bone marrow transplantation. They have been used to treat various conditions like leukemia, lymphoma, and multiple myeloma through both autologous and allogeneic hematopoietic stem cell transplants.

### 4.4. Nervous System (Brain and Spinal Cord)

Insults to the mammalian brain that cause extensive neuronal cell death are a significant challenge to treat, as the regeneration of neurons in the adult nervous system is limited. Developing novel techniques to enhance regeneration to reverse pathology, such as traumatic brain injury, is invaluable. AFSCs can differentiate into neural cells, offering the potential to treat neurological disorders [118]. Liang et al. induced a stroke injury in rats and subsequently injected hAFSCs into the affected area. They found that this treatment was neuroprotective through shafts’ angiogenic and anti-inflammatory properties [119]. AFSC therapy has also been shown to improve neurological outcomes in fetal neurological disorders such as hypoxic–ischemic encephalopathy [120].

Ultimately, AFSCs are an adequate and accessible source of stem cells that could be widely used in clinical settings to treat neurological disorders. UCBSCs can potentially treat neurological diseases like cerebral palsy. A randomized controlled trial conducted by Zarrabi et al. showed improved white matter in children diagnosed with cerebral palsy after UBSCs were injected intrathecally [121]. USCs are also being researched for neural repair and, for example, introducing exosomes from USCs aided neurogenesis after ischemic injury in rats [122]. Lastly, CSFSCs, which contain neural stem cells, show potential for spinal cord injury repair, stroke repair, and neurodegenerative diseases like Parkinson’s and Alzheimer’s [67].

### 4.5. Urinary System (Kidneys and Urinary Tract)

Over half a million people receive dialysis for end-stage renal disease in the United States [157]. Additional urinary tract pathologies such as bladder dysfunction and congenital anomalies are other significant medical challenges. Developing ways to prevent, directly treat, and potentially reverse kidney and urinary tract disease is critical. AFSCs are being researched for their renal and urothelial applications. These stem cells have been shown to preserve renal function in mouse models for progressive renal fibrosis. Samples treated with AFSCs had slower fibrotic progression and a reduced mortality [123]. It was hypothesized that this was accomplished through the modulation of the surrounding renal environment by AFSCs.

In addition, AFSCs from rats have been successfully differentiated, resembling renal cellular components [124]. Kang et al. harvested AFSCs and cultured them in a conditioned medium, eventually expressing markers (UPII, CK8, and FGF10) that are specific to the urothelium [125]. USCs also have the potential for the tissue engineering of urinary tract tissues, as well as renal regeneration. An example of USCs used for urethral tract engineering is discussed in a study by Wu et al., whereby urothelial-differentiated USCs were seeded onto a biological scaffold, which developed into a complex, multilayered structure resembling the urinary tract [126]. USCs have also been differentiated into nephron progenitor cells, which have been shown to treat chronic kidney disease by delaying renal fibrosis and enhancing regeneration, among other therapeutic mechanisms [127].

### 4.6. Reproductive System (Endometrium)

The endometrium is essential for cultivating a fertile environment for pregnancy; however, it is susceptible to damage from insults like surgical intervention, hormonal imbalance, and the formation of fibrotic adhesions [128]. Endometrial regeneration is crucial for treating endometrial disorders. This technique allows MenSCs to treat Asherman’s syndrome and other endometrial disorders. Using a specialized culture medium, Chen et al. isolated MSCs from menstrual blood and differentiated them into decidualized stromal cells.

Damaged endometrium was then treated with these cells, and a significant improvement in endometrial thickness and functional components was observed, leading to increased fertility [129]. In a separate study, research participants who were deemed infertile due to Asherman’s syndrome were treated with their MenSCs through autologous transplantation after culture expansion. Endometrial thickness also increased in some women, leading to multiple successful pregnancies [130]. Zhang et al. also investigated acellular therapeutics in the treatment of intrauterine adhesions using exomes derived from MenSCs. Fertility outcomes and general endometrial health also improved, leading to the speculation that these stem cells can also exert their effects through extracellular modalities [131]. Ultimately, using autologous cells to regenerate endometrial tissue is ideal and has proven effective at improving fertility outcomes. Despite this, further research is imperative to develop a reproducible process to use these cells widely in a clinical setting.

### 4.7. Liver

Liver repair is a vital area of research. Acute and chronic liver diseases are major causes of patient mortality worldwide [158]. However, liver transplantation is limited due to donor shortages, high costs, and transplant rejection [159]; however, cell therapy is a promising treatment.

Human umbilical cord blood mesenchymal stem cells (hUCB-MSCs), alone or combined with G-CSF, can ameliorate acute liver failure (ALF) in rats by improving liver function, reducing oxidative stress, inhibiting pro-inflammatory cytokines, and decreasing liver cell apoptosis [132]. Overexpressing VEGF165 enhances the multipotency of MSCs, promotes their homing and colonization in liver tissues, as well as contributing to liver regeneration, ultimately improving liver damage in rat models of acute liver failure [133].

Both MenSCs and their derived exosomes attenuated acute liver failure in mice by inhibiting the secretion of inflammatory cytokines [134]. MenSCs have antifibrotic effects in a liver fibrosis mouse model, improving liver function, reducing collagen deposition, and inhibiting hepatic stellate cell activation. This suggests their potential as a therapeutic approach for chronic liver diseases [135].

USC transplantation can also repair liver injury. Zhang et al. demonstrated that USCs from end-stage liver disease patients exhibit similar biological characteristics to USCs from healthy individuals. They can effectively recover liver function and improve liver tissue damage in mouse models of acute and chronic liver injury [136].

### 4.8. Skin and Wound Healing

Skin regeneration is essential for wound healing and burn treatment. AFSC-based therapies offer a promising approach for promoting scarless tissue renewal and regeneration by supporting wound healing through self-replication, differentiation, and paracrine signaling [40]. AFSCs facilitate skin wound healing and mitigate the formation of fibrotic scars by differentiating into keratinocytes and secreting paracrine signals via the TGF-β/SMAD2 pathway. They promote wound closure, remodel the extracellular matrix, reduce type I collagen, increase type III collagen, and enhance healing without affecting granulation tissue [137]. In addition, a study by Zheng et al. found that amniotic fluid-derived mesenchymal stem cell-secreted molecule (afMSC-SM) powder promotes keratinocyte self-renewal and differentiation, and can be used for wound healing and hair regrowth through hydrogel and nanogel delivery systems [138].

MenSCs combined with hyaluronic acid (HA) significantly improved diabetic wound healing in rats, as evidenced by enhanced wound contraction, tissue regeneration, collagen density, and the gene expression of TGF-β, VEGF, TNF-α, and IL-1β [139]. Dalirfardouei’s study found that exosomes secreted by MenSCs effectively improve wound healing in diabetic mice by inducing M1-M2 macrophage polarization, promoting angiogenesis, and accelerating re-epithelialization, thereby reducing scar formation [140].

### 4.9. Lungs

Lung repair is being investigated, and AFSCs are being researched for lung tissue repair. AFSCs with lung specificity can improve lung function damaged by acute lung injury during ex vivo lung perfusion and transplantation, with no adverse effects at various doses. They are associated with improved pulmonary function and reduced markers of lung injury [141]. AFSCs have also demonstrated protective and therapeutic effects against hyperoxia-induced lung fibrosis, improving pulmonary function and reducing tissue injury and inflammation [142].

MenSCs and their derived extracellular vesicles also exhibit excellent therapeutic effects in lung injury. Xu et al. conducted a multicenter, open-label, non-randomized, parallel-controlled exploratory trial involving 44 patients. They found that MenSC transplantation significantly reduced the mortality rate of severe and critically ill COVID-19 patients and improved their respiratory function [143]. Lian et al. found that extracellular vesicles secreted by menstrual blood-derived stem cells in an acute lung injury model can regulate the inflammatory response and improve lung inflammatory injury [144].

### 4.10. Intestinal System

Multiple factors cause inflammatory bowel disease (IBD), which disrupts the intestinal mucosal barrier and leads to chronic inflammation, structural changes, and dysfunction [145]. Stem cell transplantation can modulate immune cells or restore structural cells, aiding in repair or supplementation [145].

MenSCs have anti-inflammatory and immunosuppressive effects, which can be used to treat experimental colitis in mice [146]. In addition, small extracellular vesicles secreted by TNF-α-pretreated MenSCs can effectively alleviate colonic inflammation and tissue damage in a mouse colitis model [147].

AFSCs and their derived EVs can effectively improve intestinal damage in experimental necrotizing enterocolitis (NEC). AFSCs and EVs promote intestinal recovery by enhancing cellular proliferation, reducing inflammation, and ultimately regenerating normal intestinal epithelium [148].

USCs have potential as a cell-based treatment for IBD. Zhou et al. demonstrated that they reduce inflammation and alleviate IBD symptoms in rodent models by downregulating Th1/Th17 immune responses [149].

Peripheral blood stem cell transplantation is a safe and effective option for refractory Crohn’s disease that is capable of inducing remission in these patients [150].

### 4.11. Retina

Age-related retinal diseases, such as age-related macular degeneration (AMD), which is a leading cause of visual impairment in the elderly, are closely associated with the degenerative changes in retinal pigment epithelial (RPE) cells, which serve as an essential pathological basis for its development [160].

AFSCs have shown effectiveness in improving visual function in retinal degeneration (RD) rat models. This is mainly due to the paracrine effects of several growth factors that are secreted by the stem cells [151].

Conditioned medium for USCs has shown promise in significantly improving the morphology, proliferation, and apoptosis of aging RPE cells by regulating multiple gene networks, thereby slowing the aging process [152]. In addition to RPE cells, retinal ganglion and nerve cells are essential retina components. Shi et al. demonstrated that exosomes derived from USCs can reduce apoptosis and enhance cell viability and proliferation in aging retinal ganglion cells [153].

## 5. Summary

Body fluid-derived stem cells offer significant advantages in regenerative medicine due to their unique properties and ease of access. Isolated from various bodily fluids through non-invasive or minimally invasive methods [57], BFSCs represent a readily available source compared to traditional stem cell sources from tissues like bone marrow. These stem cells share key characteristics with the MSCs found in bone marrow [10]. They can also self-renew and differentiate into various cell types, including bone, cartilage, muscle, and nerve cells. In addition, BFSCs exhibit immunomodulatory properties, making them potentially suitable for treating autoimmune diseases. Immunomodulation is critical in tissue regeneration by influencing the immune response to promote healing and repair. These properties foster a balanced immune response that reduces inflammation, enhances tissue repair, prevents rejection, and promotes angiogenesis, which are all essential for effective tissue regeneration. This combination of advantages, non-invasive collection, abundant source, immunomodulatory properties, and regenerative potential positions BFSCs as being appropriate for exciting applications in regenerative medicine.

While BFSCs offer advantages like less-invasive collection and unique properties, BFSC research remains in its early stages, and their journey to widespread clinical and research use is ongoing. Overcoming challenges related to efficient isolation, characterization, scalability, safety, efficacy, regulation, and continued research investment will be crucial to realizing their full potential in regenerative medicine.

In summary, extracellular body fluids such as urine, blood, breast milk, menstrual fluid, umbilical cord blood, and specialized fluids like amniotic fluid and cerebrospinal fluid (CSF) are emerging areas of exploration for stem cell sourcing. These fluids offer advantages, including non-invasive or minimally invasive collection methods and potentially unique stem cell populations. However, current research is primarily focused on identifying and characterizing these populations. While their applications in disease modeling, drug testing, and therapeutics show great promise, these areas remain exploratory. Several research areas need to be prioritized to overcome common barriers to implementing BFSCs, which were addressed in this review. First, the standardization of isolation, expansion, and storage protocols must be developed to ensure consistent therapeutic efficacy and safety. Second, personalized therapies should be investigated to reduce immunogenicity and enhance therapeutic potential based on individual requirements. Third, BFSCs should be integrated with other novel technologies like genetic engineering, EV-based cargo delivery, and biomaterials. A focused approach on these aspects will make utilizing BFSCs in clinical settings much more feasible. Thus, the potential of body fluids as a future source of stem cells is promising, representing an exciting avenue for further research in regenerative medicine.

## Figures and Tables

**Figure 1 ijms-26-04382-f001:**
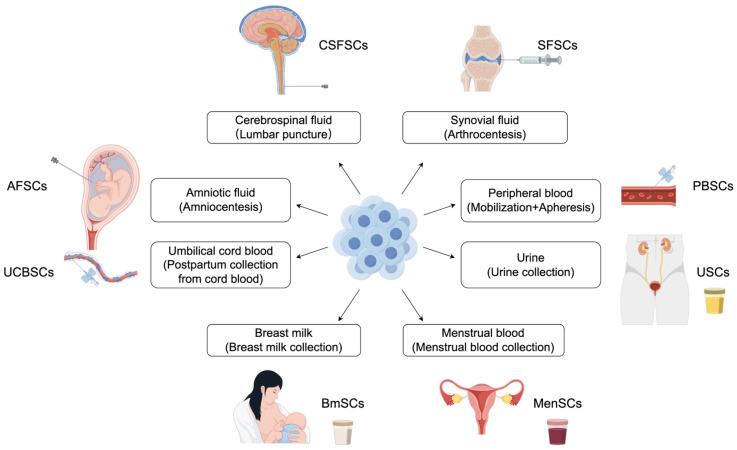
Body fluid as a reservoir of stem and progenitor Cells (made using Figdraw). This schematic illustrates the potential of various body fluids, including cerebrospinal fluid, synovial fluid, urine, peripheral blood, menstrual blood, breast milk, umbilical cord blood, and amniotic fluid, as sources for isolating mesenchymal stem cells and other valuable progenitor cell populations.

**Table 1 ijms-26-04382-t001:** Classification of stem cells—potency, cell type, and origin.

Classification Basis	Description	Examples
Potency	Ability of a stem cell to give rise to various cell types	- Totipotent: differentiate into all cell types of an organism (only present in the fertilized egg or zygote) [11,12]. - Pluripotent: differentiate into almost any cell type found in the body (excluding those for fetal development) [11,12]. - Multipotent: differentiate into a limited number of closely related cell types [11,12]. - Oligopotent: differentiate into just a few specific cell types [11,12]. - Unipotent: only differentiate into one specific cell type [11,12].
Development Stage	When the stem cells arise during an organism’s development	- ESCs: derived from the inner cell mass of an early-stage embryo [13]. - FSCs: isolated after 9 weeks of gestation from the fetus or extra-embryonic tissues [14,15]. - iPSCs: artificially created in the lab by reprogramming adult cells to an embryonic-like state [16]. - Perinatal stem cells: found in tissues associated with pregnancy, which are obtained immediately after the baby is born, e.g., umbilical cord blood, umbilical cord tissue, placenta, and amniotic fluid [17,18]. - Adult stem cells: Found in various tissues throughout the body after birth [19].
Cell Types	Focuses on the specific lineages or tissues that the stem cells can differentiate into	- Epithelial stem cells: located in the skin, lining of organs, etc., differentiate into various epithelial cell types [20,21]. - MSCs: located in connective tissues and can differentiate into bone, cartilage, fat, muscle, and endothelial cells [22]. - HSCs: Found in bone marrow and can differentiate into all types of blood cells [23]. - Neurogenic stem cells: reside in specific regions of the central nervous system and can differentiate into new neurons and glial cells [24].
Origin	Source of the stem cells within the body	- TSCs: reside within specific tissues (bone marrow, fat, muscle, liver, kidney, skin, and hair folic) [25,26]. - BFSCs: extracted from fluids containing cells that have migrated from their tissue niches (amniotic fluid, urine, breast milk, menstrual blood, peripheral blood, umbilical cord blood, cerebrospinal fluid, salivary gland fluid, and synovial fluid) [10,27,28].

Abbreviations—BFSCs: body fluid-derived stem cells; ESCs: embryonic stem cells; FSCs: fetal stem cells; HSCs: hematopoietic stem cells; iPSCs: induced pluripotent stem cells; MSCs: mesenchymal stem cells; TSCs: tissue-derived stem cells.

**Table 2 ijms-26-04382-t002:** Classification of body fluid stem cells by cell markers, advantages, and limitations.

Source	Key Markers	Advantages	Limitations
**Fetal or Neonatal Stem Cells**			
Amniotic Fluid	CD73, CD90, CD105, SSEA4, c-Kit, TRA-1-60, TRA-1-81, PSG5, EMX-2, and EVR-3 [38]	Differentiation potential, immunomodulatory properties [39]; lower rate of senescence [33]; minimal ethical concern [40]	Heterogenous population [31]; requires specific gestational timing; risk associated with amniocentesis
Umbilical Cord Blood	CD34, CD45, and CD117 (c-Kit) [41]	High and preserved differentiation capacity [42]; low immunogenicity [43]	Limited cell quantity per sample; mostly hematopoietic cells (a few MSCs) [44]
**Adult Stem Cells**			
Peripheral Blood	CD34 and CD90 [45]	Ease of collection [46,47,48]; faster engraftment after transplantation [49]	Low HSC numbers; higher incidence of GVHDs [49]; requires mobilization agent or growth factor injection [46,47,48]
Menstrual Blood	-ASC markers: CD29, CD44, CD73, CD90, and CD105 [50]; -ESC markers: OCT-4, SOX2, and SSEA-4 [50]	High proliferative potential; expression of adult and embryonic markers [51]; non-invasive alternative [52]	Affected by donor age, hormonal status, and contraceptive use; optimal isolation and sterilization techniques are yet to be established [52]
Urine	-MSC markers: CD73, CD90, and CD105 [53]; -RPC markers: SIX2, CITED1, WT1, CD24, and CD106 [54]	MSC-like properties [55]; high telomerase activity and karyotype levels after in vitro expansion; immunomodulating properties [56]; low tumorgenicity [57]; non-invasive [54]	Vary Isolation efficiency between individuals; regenerative ability reduced in USCs from aged donors or diabetic nephropathy [58]; potential for contamination, particularly in collecting urine samples from females [56]
Synovial Fluid	MSC-like markers: CD73, CD90, CD105, and CD44 [59]	Chondrogenic differentiation potential; immunomodulatory properties; anti-inflammatory effects [60]	Lower proliferation potential; limited fluid volume [61]; quality and quantity affected by age and joint disease
Breast Milk	-MSC markers: CD90, CD44, CD271, and CD146 [62]; -ESC markers in subpopulation: TRA 60-1, Oct4, Nanog, and Sox2 [62]	ESC gene expression [63]; multilineage differentiation [64]; non-invasive collection [64]	Heterogenicity of cell population [65]; limited cell number and suboptimal culture conditions [65]; cell yield may vary between individuals and lactation stages
Cerebro- spinal Fluid	Neural progenitor markers: TBR2, CD15, and SOX2 [66]	Treats neurodegenerative diseases and neuronal ischemic injury [67]	Challenging to harvest and low cell yield [66]; differentiation potential restricted primarily to neural lineages; potential for tumor formation (gliomas)

Abbreviations—GVHD: graft vs. host disease; ASC: adult stem cell; ESC: embryonic stem cell; MSC: mesenchymal stem cell; RPC: renal progenitor cell; NSC: neuronal stem cell.

**Table 3 ijms-26-04382-t003:** Body fluid-derived stem cell applications across systems.

Tissue/Organ System	BFSCs with Applications	References
Musculoskeletal	AFSCs, PBSCs (MSCs), MenSCs, USCs, SFSCs	[61,76,103,104,105,106,107,108,109,110,111]
Cardiovascular	AFSCs, PBSCs (HSCs/MSCs), UCBSCs, MenSCs	[112,113,114,115,116]
Hematopoietic	PBSCs (HSCs), UCBSCs	[43,49,117]
Nervous	AFSCs, UCBSCs, USCs, CSFSCs	[67,118,119,120,121,122]
Urinary	AFSCs, USCs	[123,124,125,126,127]
Reproductive (Endometrium)	MenSCs	[128,129,130,131]
Liver	MenSCs, UBCSCs, USCs	[132,133,134,135,136]
Skin/Wound Healing	AFSCs, MenSCs	[40,137,138,139,140]
Lungs	AFSCs, MenSCs	[141,142,143,144]
Intestinal	MenSCs, AFSCs, USCs	[145,146,147,148,149,150]
Retina	AFSCs, USCs (conditioned medium)	[151,152,153]

Abbreviations—AFSCs: amniotic fluid-derived stem cells; PBSCs: peripheral blood-derived stem cells; MSCs: mesenchymal stem cells; MenSCs: menstrual blood-derived stem cells; USCs: urine-derived stem cells; SFSCs: synovial fluid-derived stem cells; HSCs: hematopoietic stem cells; UCBSCs: umbilical cord-derived stem cells; CSFSCs: cerobrospinal fluid-derived stem cells.

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
