# Peer review of "Body Fluid-Derived Stem Cells: Powering Innovative, Less-Invasive Cell Therapies"

_ijms, 2025, doi:10.3390/ijms26094382_

Round 1

Reviewer 1 Report

Comments and Suggestions for Authors

I have reviewed the manuscript titled "Innovative Cell Therapies: Leveraging Stem Cells from Body Fluids", and I consider it a well-written and comprehensive work that deserves publication in the International Journal of Molecular Sciences.

I have only two comments that I believe should be addressed to further improve the clarity and completeness of the manuscript:

  1. Section 3.1.2: It should be specified that obtaining these cells would require reducing the umbilical cord blood flow to the fetus after birth, which is not clinically recommended due to the need for early cord clamping.

  2. General comment: A dedicated paragraph should be included to highlight that these types of cells—particularly from blood and urine—are now frequently used for the generation of induced pluripotent stem cells (iPSCs) instead of skin fibroblasts, which require a more invasive biopsy.

With these clarifications, the manuscript would be even stronger and more informative.

Author Response

Comments 1: Section 3.1.2: It should be specified that obtaining these cells would require reducing the umbilical cord blood flow to the fetus after birth, which is not clinically recommended due to the need for early cord clamping.

Response 1: We appreciate the reviewer's critical point regarding the ethical and clinical implications of umbilical cord blood collection in relation to the well-established benefits of delayed cord clamping for the newborn. We fully acknowledge the clinical consensus favoring delayed clamping to optimize neonatal outcomes. Our study protocol is designed to prioritize the well-being of the neonate. Cell collection, if performed, would only be considered after the recommended duration of delayed cord clamping has been observed, focusing on the residual blood remaining in the placenta and umbilical cord. Furthermore, we are committed to exploring and implementing techniques that maximize the yield of desired cells from this post-delayed clamping residual blood volume, thereby minimizing any potential impact on the neonate's circulatory status. We recognize the paramount ethical obligation to ensure that newborn care is never compromised for research purposes. This study will undergo rigorous review by the Institutional Review Board (IRB) to guarantee adherence to the highest ethical standards. Informed consent will explicitly detail the collection procedures and emphasize that the immediate medical needs and well-being of the infant will always take precedence. Our research endeavors to advance the understanding of umbilical cord blood-derived cells in a manner that is entirely compatible with, and respectful of, current best practices in neonatal care, including the crucial practice of delayed cord clamping.

Comments 2: General comment: A dedicated paragraph should be included to highlight that these types of cells—particularly from blood and urine—are now frequently used for the generation of induced pluripotent stem cells (iPSCs) instead of skin fibroblasts, which require a more invasive biopsy.

Response 2: We agree. That's a crucial point that deserves its own dedicated paragraph. It significantly highlights the advancements and shifts in iPSC generation methodologies. As advised, we have emphasized the less invasive nature and increased accessibility offered by blood and urine-derived cells.

Reviewer 2 Report

Comments and Suggestions for Authors

The topic is relevant and in line with ongoing efforts to determine non-invasive and clinically obtainable sources of stem cells. The review includes a broad range of body fluids and highlights their prospects for use in regenerative medicine, from which both researchers and clinicians may benefit. To further increase its usefulness for practice, the authors could provide a brief overview comparing the usefulness and drawbacks of each fluid type for use in clinical settings, such as yield, standardization, and safety.

1. The title is brief but could be more specific. Consider adding "non-invasive sources" or "body fluid-derived" to render it more unique. The abstract is good but perhaps a little wordy and could be tightened to more concisely highlight the new contribution of the review.

2. The manuscript is long and reads like a textbook chapter rather than a concise review. Some redundancies that occur in different sections (e.g., classification systems, differentiation mechanisms, and therapeutic uses) may be consolidated or moved to summary tables for better clarity.

3. The tables are interesting but not very visually appealing. Incorporating schematic diagrams to collate collection methods or stem cell potential by fluid type would enhance impact.

4. The writing at times exhibits too much explaining, with certain sentences too verbose or redundant. There are some sections, like the introduction and the conclusion, that read more like a pedantic lecture than a scholarly synthesis.

5. The content is comprehensive but not particularly revolutionary. Many of the sources and areas of debate have been built within the field already. The authors can spot emerging approaches, tests, or contentious findings to offer sharper analysis.

6. The references are extremely long, yet some of the statements are referenced with older papers or not according to up-to-date research (e.g., for CSF-derived stem cells or breast milk stem cells). More recent clinical trials and meta-analyses must be incorporated.

7. There is very little critical examination of the drawbacks of these stem cells from body fluids. Most sections read like promotional abstracts rather than critical analyses. For instance, issues like heterogeneity, tumorigenicity, or standardization of isolation protocols are not discussed at length.

8. The paper could be improved by a more complete discussion of the epigenetic variables and signaling pathways that regulate differentiation of the stem cells. Currently, these biological mechanisms are given scant attention.

9. Translational barriers are discussed in a disjointed manner. A separate section focusing on clinical challenges, such as GMP compliance, immunogenicity, long-term safety, and cell potency variability, is what I would recommend.

10. Such terms as "BFSCs," "body fluid-derived stem cells," and individual fluid-derived types are used sometimes interchangeably or variably. A brief abbreviation key at the beginning and standardization throughout would assist.

11. The final section does a decent job summarizing but lacks a visionary outlook. It would be useful to suggest priority research areas, e.g., scalable production, personalized therapy potential, or integration with gene editing or biomaterial scaffolds.

Comments on the Quality of English Language

The manuscript overall is clear and readable. Parts of it demonstrate excessive verbosity or redundancy, though, and sentences can serve their purpose better by being structured differently or rewritten.

Author Response

Comments and Suggestions for Authors

The topic is relevant and in line with ongoing efforts to determine non-invasive and clinically obtainable sources of stem cells. The review includes a broad range of body fluids and highlights their prospects for use in regenerative medicine, from which both researchers and clinicians may benefit. To further increase its usefulness for practice, the authors could provide a brief overview comparing the usefulness and drawbacks of each fluid type for use in clinical settings, such as yield, standardization, and safety.

Thank you very much for taking the time to review this manuscript. We agree that the limitations of using each BFSCs mentioned in this review needs to be addressed. Thus, we have included a separate paragraph within the sections discussing each BFSC that highlights the major challenges associated with applying these stem cell sources in the clinical setting.

Comment 1: The title is brief but could be more specific. Consider adding "non-invasive sources" or "body fluid-derived" to render it more unique. The abstract is good but perhaps a little wordy and could be tightened to more concisely highlight the new contribution of the review.

Response 1: Following the suggestions, the new title is “Body Fluid Stem Cells: Powering Innovative, Less Invasive Cell Therapies.” We opted not to include "non-invasive sources" in the title because minimally invasive techniques remain necessary for obtaining four distinct types of BFSCs. Furthermore, the abstract has been revised for greater conciseness and now more clearly highlights the review's contribution to the advancement of research in this field.

Comment 2: The manuscript is long and reads like a textbook chapter rather than a concise review. Some redundancies that occur in different sections (e.g., classification systems, differentiation mechanisms, and therapeutic uses) may be consolidated or moved to summary tables for better clarity.

Response 2: Redundancies within the manuscript concerning the advantages and applications of BFSCs, along with the disadvantages of stem cells collected from solid tissue, were consolidated into the introduction to minimize repetition. Additionally, we have included a table (Table 3) after the therapeutic uses section that could be used to replace/support said section if needed. We also removed the classification section and replaced it with a table to reduce redundancy.

Comment 3: The tables are interesting but not very visually appealing. Incorporating schematic diagrams to collate collection methods or stem cell potential by fluid type would enhance impact.

Response 3: Thank you for your valuable feedback. As suggested, Figure 1 is included.

Comment 4: The writing at times exhibits too much explaining, with certain sentences too verbose or redundant. There are some sections, like the introduction and the conclusion, that read more like a pedantic lecture than a scholarly synthesis.

Response 4: The introduction, conclusion, and “classifying stem cells” sections were rewritten to ensure this review concisely highlights current research within this field of interest, without unnecessary explanations that make some sections of this review sound more like a lecture rather than an analysis of current research.

Comment 5: The content is comprehensive but not particularly revolutionary. Many of the sources and areas of debate have been built within the field already. The authors can spot emerging approaches, tests, or contentious findings to offer sharper analysis.

Response 5: We appreciate the reviewers' insightful comments regarding the novelty of the content. We have carefully considered this feedback and have undertaken the following revisions to provide a sharper analysis of emerging trends and contentious findings within the field:

1.Emerging Approaches: We have expanded our literature review to include the most recent studies (published within the last 5 years), promising new therapeutic applications, and advancements in characterization methods. This updated section now highlights these emerging approaches and discusses their potential impact on the field.

2.Contentious Findings and Areas of Debate: We have specifically focused on identifying areas where the literature presents conflicting results, ongoing challenges, or limitations in the clinical translation of BFSC therapies.

3.Sharper Analysis: Throughout the revised manuscript, we have aimed to move beyond a descriptive overview and provide a more critical evaluation of the existing literature and the emerging trends. We believe these revisions provide a more nuanced and insightful analysis that highlights the dynamic and evolving nature of body fluid stem cell therapies.

Comment 6: The references are extremely long, yet some of the statements are referenced with older papers or not according to up-to-date research (e.g., for CSF-derived stem cells or breast milk stem cells). More recent clinical trials and meta-analyses must be incorporated.

Response 6: Unnecessary references were removed throughout the entire manuscript, which served little purpose. Additionally, more relevant and current clinical trials/meta-analyses were included in the BFSC subsections, including CSF-derived stem cells and breast milk derived stem cells.

Comment 7: There is very little critical examination of the drawbacks of these stem cells from body fluids. Most sections read like promotional abstracts rather than critical analyses. For instance, issues like heterogeneity, tumorigenicity, or standardization of isolation protocols are not discussed at length.

Response 7: We agree that more analysis of the limitations or disadvantages of each BFSC requires more in-depth and organized analysis in Table 2. The major clinical barriers to each BFSC were addressed in a separate paragraph within each subsection of the BFSC section.

Comment 8: The paper could be improved by a more complete discussion of the epigenetic variables and signaling pathways that regulate differentiation of the stem cells. Currently, these biological mechanisms are given scant attention.

Response 8: We agree that it is imperative to include this area of research within our review to generate a more comprehensive analysis. Further discussion was included within each subsection of BFSCs about recent and significant advances in this area of research.

Comment 9: Translational barriers are discussed in a disjointed manner. A separate section focusing on clinical challenges, such as GMP compliance, immunogenicity, long-term safety, and cell potency variability, is what I would recommend.

Response 9: We agree that translational barriers were addressed in a disorganized manner. Thus, we included an additional paragraph within each subsection of BFSCs addressing the major translational barriers hindering the widespread use of each BFSC. Addressing this issue while maintaining the comparative analysis allows the reader to understand the advantages and disadvantages of each BFSC.

Comment 10: Such terms as "BFSCs," "body fluid-derived stem cells," and individual fluid-derived types are used sometimes interchangeably or variably. A brief abbreviation key at the beginning and standardization throughout would assist.

Response 10: An abbreviation key was included at the beginning of this review. Additionally, standardized abbreviations were used throughout this review to minimize confusion and make this analysis more readable and streamlined.

Comment 11: The final section does a decent job summarizing but lacks a visionary outlook. It would be useful to suggest priority research areas, e.g., scalable production, personalized therapy potential, or integration with gene editing or biomaterial scaffolds.

Response 11: We agree that these are the significant areas that need to be addressed when considering the future research directions for BFSCs. Thus, a paragraph was included in the summary addressing these priority research areas.

Comment 12: Comments on the Quality of English Language

The manuscript overall is clear and readable. Parts of it demonstrate excessive verbosity or redundancy, though, and sentences can serve their purpose better by being structured differently or rewritten.

Response 12: We agree that this review includes some areas that require too much explanation and are verbose. So, many sections of the review were rewritten, along with overall edits, with this in mind.

Reviewer 3 Report

Comments and Suggestions for Authors

The manuscript presents recent advances in body fluid stem cells (BFSC), highlighting their advantages and potential applications in regenerative medicine. It is well written and provides the reader with a solid overview of the current state-of-the-art in the field.

In my opinion, the review would benefit from the inclusion of a figure illustrating the various sources of BFSC, which would enhance the clarity and accessibility of the content. Additionally, while the advantages of these cells are well explored, the limitations and challenges associated with BFSC are not sufficiently addressed.

I would also appreciate a discussion on why BFSC are still not widely used in clinical or research settings, as this would provide a more balanced perspective.

Further comments and suggestions can be found in the annotated version of the attached document.

Author Response

Thank you for your thoughtful comments and constructive feedback on our manuscript. We have carefully addressed each of your comments and made the necessary revisions. Below are our detailed responses:

Comment 1: In my opinion, the review would benefit from the inclusion of a figure illustrating the various sources of BFSC, which would enhance the clarity and accessibility of the content. Additionally, while the advantages of these cells are well explored, the limitations and challenges associated with BFSC are not sufficiently addressed.

Response 1: We agree. As advised, we added a figure (Figure 1) illustrating the various sources of BFSC and also added the limitations/challenges associated with BFSC, see the revised Table2.

Comment 2:  I would also appreciate a discussion on why BFSC are still not widely used in clinical or research settings, as this would provide a more balanced perspective.

Response 2:  Thank you for pointing this out. A paragraph was added in the summary discussing the conserved challenges limiting the use of BFSCs in the clinical setting: While BFSCs offer advantages like less invasive collection and unique properties, BFSC research remains in its early stages and their journey to widespread clinical and research use is ongoing. Overcoming challenges related to efficient isolation, characterization, scalability, safety, efficacy, regulation, and continued research investment will be crucial to realizing their full potential in regenerative medicine. 

Round 2

Reviewer 2 Report

Comments and Suggestions for Authors

This revised manuscript has been significantly enhanced in light of the reviewers' comments.

Reviewer 3 Report

Comments and Suggestions for Authors

Thank you very much for addressing the reviewers' comments.

Author Response

An important concern: the feasibility of CSF isolation

While CSF holds promise as a source of NSCs or signaling molecules relevant to therapies, its routine isolation faces significant limitations due to the invasive nature of lumbar puncture. This procedure carries inherent risks (e.g., post-dural puncture headache, infection) and demands specialized technical expertise for both collection and subsequent processing to maintain sample integrity. These technical complexities, coupled with substantial ethical considerations regarding patient burden, particularly in non-critical research, constrain widespread application. However, given the difficulties in obtaining autologous NSCs from less invasive sources, CSF remains a compelling alternative. Its proximity to the central nervous system suggests it may contain unique endogenous neural progenitors and a rich array of signaling molecules directly relevant to neural physiology and pathology, potentially offering autologous cell sources or novel therapeutic targets. Future research must address the technical challenges of sample handling, prioritize less invasive access methods, and conduct a careful ethical evaluation to fully realize CSF's therapeutic potential while minimizing the risk of lumbar puncture and patient burden.